# Research on the Influence Mechanism and Configuration Path of Network Relationship Characteristics on SMEs' Innovation—The Mediating Effect of Supply Chain Dynamic Capability and the Moderating Effect of Geographical Proximity

Hongxiong Yang and Wanru Ren *

School of Management, Tianjin University of Technology, Tianjin 300384, China;
yanghongxiong@email.tjut.edu.cn
* Correspondence: rwr961006@163.com

**Abstract:** Innovation is the continuous source of power for the survival and development of SMEs, but the complexity of innovation and the limitation of resources make SMEs trapped in the dilemma of "innovation difficulty". A moderated mediating model was constructed based on social network theory and resource view, and fuzzy set qualitative comparative analysis (fsQCA) was used to empirically study the influence mechanism between network relationship characteristics and SMEs' innovation and the configuration path to achieve SMEs' high innovation performance. The results show that the characteristics of network relationships positively affect the innovation performance of SMEs. Supply chain dynamic capability plays an intermediary role between network relationships and SMEs' innovation relationships. Different geographical proximity levels of the supply chain lead to different coordination interaction and knowledge sharing efficiency between upstream and downstream, which not only positively moderates the relationship between supply chain dynamic capability and SMEs' innovation performance, but also moderates its mediating effect. Furthermore, the fs QCA analysis results show three configurations for SMEs' high innovation performance based on the characteristics of network relations: geographic proximity regulating type, network relationship dominant type, and dynamic coordination and integration type.

**Keywords:** network relationship; supply chain dynamic capability; geographical proximity; SMEs innovation; fs QCA



## 1. Introduction

In response to the development of international unilateralism and the intensification of international environmental turmoil, China has proposed a dual-cycle development strategy, which requires the promotion of innovation and development based on promoting the modernization of the industrial chain and supply chain. Among them, SMEs are one of the mainstays of Chinese economic development, and stimulating their innovative vitality is the key to building a new dual-cycle pattern [1]. However, on the one hand, due to increased environmental uncertainty, SMEs are facing greater innovation risks; on the other hand, they are limited by their scale, resulting in a lack of a resource base for SMEs' innovation [2]. As an important social resource for enterprises, network relationships can provide heterogeneous resources and opportunities for the development of SMEs [3,4]. In the supply chain network, more and more SMEs choose to cooperate and share with upstream and downstream partners to obtain access to resources in the supply chain [5,6]. Take computer manufacturers such as Lenovo and Dell as examples. After Microsoft independently developed a new hardware product—the Surface 2-in-1 tablet—in 2012, they cooperated with Microsoft to promote the development of their traditional products in the direction of becoming thinner and lighter. Therefore, exploring the influence mechanism of network relationships on SME innovation is not only the focus of implementing the

dual-cycle development strategy, but also an inevitable requirement for SMEs to achieve sustainable development.

Although the current research on the influence of network relationships on the innovation of SMEs has been confirmed by some scholars, the specific influence mechanism still needs to be further explored. Dynamic capability refers to the ability of an enterprise to quickly integrate network resources according to changes in the external environment and reconfigure according to demand [7]. In the supply chain, the main performance is that upstream and downstream companies can quickly adapt to environmental changes, and effectively manage network relationships and internal and external resources according to market demand, thereby promoting corporate innovation [8–10]. Therefore, the role of supply chain dynamic capabilities in SME innovation needs to be explored.

Companies with close geographical distances generally have a high degree of similarity in social, market, and institutional environments, which makes it easier and faster for companies to acquire new knowledge [11,12]. At the same time, the frequency and efficiency of the interaction between upstream and downstream enterprises in the supply chain will also be affected by geographic spatial distance, thereby affecting the output of innovation results. Therefore, it is necessary to further analyze whether the geographical proximity of the supply chain will affect the innovation of SMEs.

Therefore, this research aims to solve the following questions: will the network relationship of SMEs affect innovation performance, and what role does the dynamic capability of the supply chain play? Secondly, based on the different levels of geographic proximity of the supply chain, are there different configurations that affect the innovation of SMEs? This study will construct a mediation adjustment model based on the mediation effect of supply chain dynamic capabilities and the adjustment effect of geographic proximity. First, through theoretical analysis and literature review, the research hypothesis model is constructed. Second, the research hypothesis proposed is verified through regression analysis. Finally, through the fuzzy set qualitative comparative analysis (fsQCA), we explore the configuration of factors affecting the innovation of SMEs and analyze the driving mode of each configuration. The research results will provide relevant suggestions for resource-constrained SMEs to achieve sustainable development. The research model is shown in Figure 1.

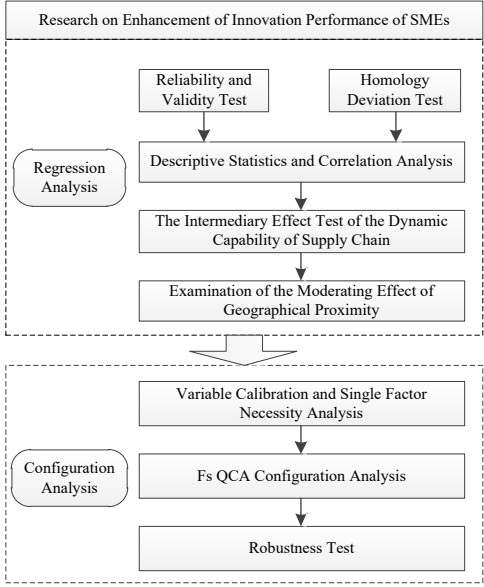

**Figure 1.** Research Model.

## 2. Theoretical Analysis and Research Hypotheses

### 2.1. Characteristics of Network Relationship and SMEs' Innovation

The characteristics of network relationships mainly refer to two levels: network relationship strength and network relationship quality. To cope with the fierce market competition and the increasing instability of the environment, more and more SMEs choose to establish a network relationship with other subjects in the supply chain to realize resource sharing to overcome the limitations of resource shortage on enterprise innovation behavior. A network relationship is the sum of social relations established by organizations in resource exchange and information exchange and is the condition and basis for organizations to search for external knowledge. According to social network theory and resource view, resource demand makes organizations dependent on external networks, and the network relationship established thus becomes an essential way for enterprises to obtain innovative resources and improve competitiveness [13]. Network relationship characteristics usually describe the nature of the relationship established between an organization and other subjects, including strength, quality, and stability [14,15]. Since relationship quality usually includes stability, the characteristics of network relationships are mainly described from two aspects: relationship strength and relationship quality.

The intensity of network relationships refers to the frequency of information exchange and resource sharing among network subjects and the closeness of the mutual relationship. Granovetter proposed dividing network relations into strong and weak ones [14]. Strong links could increase the mutual trust among various subjects, facilitate resource replacement, and thus obtain more innovation opportunities. In research on relationship strength and SME innovation, He et al. believe that the stronger the network relationship, the faster the speed of information exchange and resource replacement between enterprises and other entities, to improve the R&D capability of enterprises [16]. Huang et al. believe that relationship strength can effectively promote knowledge transfer among technology-based SMEs and form the primary resources to support enterprise innovation [17]. In conclusion, it can be concluded that a solid network relationship can improve the interaction frequency and the speed of resource replacement between organizations and help SMEs to quickly dig out the resources favorable to their innovation activities and carry out innovation activities.

Network relationship quality is the comprehensive judgment and evaluation of the trust degree and interaction effect between network subjects [18]. As for the relationship between relationship quality and SME innovation, GulatiR and Gargiulo pointed out that good relationship quality can enhance the close connection between enterprises on network nodes, promote enterprises to learn new knowledge and absorb new information, and thus improve enterprise innovation performance [19]. Zhan and Wang believed that network relationship quality affects enterprise innovation performance through knowledge acquisition ability, and good relationship quality is conducive to the mutual trust degree of member firms in collaborative innovation, which facilitates the effective acquisition of innovation resources [20]. Xie et al. believe that enhancing trust among members is conducive to enterprises' obtaining valuable resources from them, improving their confidence in the effect of behaviors, and facilitating enterprises' technological innovation behaviors [21]. To sum up, the higher the quality of the network relationship, the stronger the trust between various entities, and enterprises can acquire valuable information and resources from the network more quickly.

Accordingly, the following research hypothesis is proposed:

**Hypothesis (H1).** *The characteristics of a network relationship have a positive effect on the improvement of SMEs' innovation performance.*

### 2.2. The Mediating Role of Supply Chain Dynamic Capability

At present, under the background of "double-cycle" economic development, active participation in the supply chain will become the development trend [1]. To achieve sustainable development, SMEs must enhance their competitiveness through the dynamic

interaction between the upstream and downstream players of the supply chain. The dynamic capability of the supply chain refers to the ability to effectively promote the cooperation and complementarity, interactive learning, internal and external resource adjustment, and reorganization of enterprises in the supply chain network to quickly respond to the complex changes in the external environment, which mainly includes supply chain coordination capability, learning and absorption capability, and resource integration and reconstruction capability [22]. Existing studies show that the deepening of vertical coordination between supply chains will strengthen information sharing among nodal enterprises, promote resource homogenization, and effectively improve the innovation performance of SMEs in the growth stage [23]. Lin and Zhang found through empirical research that learning absorptive capacity, including organizational learning complementary effect, positively impacts innovative firms' dual innovation [24]. According to Li et al., enterprises can effectively promote technological and product innovation by rapidly integrating and allocating internal and external resources for strategic transformation according to changes in the market environment and demand [25]. Based on this judgment, supply chain dynamic capability may positively promote SMEs' innovation performance.

As an essential relationship capital of enterprises, the network relationship can promote the release, acquisition, integration, and reconstruction of resources in the supply chain and is the basis of dynamic integration of internal and external resources in the supply chain. Chi et al. found that the stronger the relationship, the higher the degree of resource sharing among network subjects, which is more conducive to mutual coordination, knowledge absorption, and resource integration among supply chain member enterprises [26]. Wang et al. believe that frequent communication and contact between network subjects is conducive to supply nodal chain enterprises to discuss challenging problems in innovation and research jointly, promote internal and external knowledge integration and transformation, and improve the dynamic capability of the supply chain [27]. Yli-renko and Sapienza believe that good network relationship quality can provide trust guarantee for the quality and efficiency of resource sharing among various subjects, improve the breadth and depth of interaction and communication, strengthen the partnership between enterprises, and improve the efficiency of knowledge absorption and transformation of the whole supply chain through knowledge sharing [28]. Based on this judgment, the strength and quality of network relationships may positively affect the dynamic capability of the supply chain.

Accordingly, the following research hypothesis is proposed:

**Hypothesis (H2).** *Supply chain dynamic capability plays a mediating role in the relationship between network relationship characteristics and SMEs' innovation performance.*

*2.3. The Moderating Effect of Geographical Proximity of Supply Chain*

The geographical proximity of the supply chain can also be understood as the spatial proximity or spatial distance between nodal enterprises. The closer the spatial distance between network entities, the more helpful it is to improve the frequency and efficiency of communication and interaction, reduce the information asymmetry between enterprises, and thus reduce the innovation risk of enterprises [29]. Existing studies have found that geographical proximity is conducive to the coordination, interaction, and knowledge sharing between the upstream and downstream of the supply chain, enabling enterprises to understand the market demand quickly, acquire technical knowledge and make timely adjustments to avoid ineffective innovation [30]. At the same time, the closer the space distance between enterprises in supply chain nodes is, the more conducive to knowledge spillover and interactive learning between enterprises, reducing the cost of learning new technologies and promoting innovation and development [31]. Therefore, the closer the distance between the supply chain nodes, the more coordinated supply chain capabilities, absorption capacity, and resource integration capacity, which is conducive to the development of enterprise innovation activities, which is more conducive to the development of enterprise innovation activities. Because of the complicated network relationship, the proximity of the supply chain not only adjusts the dynamic capability and the relationship

between the enterprise innovation, but also affects the ability of the supply chain dynamic mediation effect, when the space distance between the enterprise and supply chain on other subjects and more recently, the dynamic capability of the supply chain in the network relation intensity/intermediary role between the quality and the innovation performance of SMEs.

Accordingly, the following research hypotheses are proposed:

**Hypothesis (H3).** *Supply chain geographic proximity positively moderates the relationship between supply chain dynamic capability and SMEs' innovation performance.*

**Hypothesis (H4).** *Supply chain geographic proximity positively moderates the mediating role of supply chain dynamic capability between network relationship and SMEs' innovation performance.*

The research hypotheses of this paper are shown in Table 1.

**Table 1.** Research Hypotheses.

| Hypotheses | Hypothetical Content |
|---|---|
| H1 | The characteristics of network relationships have a positive effect on the improvement of SMEs' innovation performance |
| H2 | The supply chain dynamic capability plays a mediating role in the relationship between network relationship characteristics and SMEs' innovation performance |
| H3 | Supply chain geographic proximity positively moderates the relationship between supply chain dynamic capability and SMEs' innovation performance |
| H4 | Supply chain geographic proximity positively moderates the mediating role of supply chain dynamic capability between network relationship and SMEs' innovation performance |

## 3. Research Design

### 3.1. Data Sources and Sample Characteristics

SMEs refer to enterprises with a relatively small scale (usually with no more than 2000 personnel), including medium-sized enterprises, small enterprises, and microenterprises. Due to their relatively limited human, financial, material, and other resources, SMEs often invest their limited resources in small markets ignored by large enterprises to gain a firm foothold in the market competition by improving product quality and production efficiency. The research sample enterprises are determined according to the Regulations on classification standards for SMEs, and the relevant data are obtained using a questionnaire. The research areas include Beijing, Tianjin, Hebei, Shanxi, Guangdong, Jiangsu, and other areas where SMEs gather. The respondents are SMEs that have been established and officially operated for more than 3 years. The questionnaire is filled in by managers who have worked in the enterprise for two years and are familiar with the overall operation of the enterprise. In this survey, a total of 300 questionnaires were issued, and 203 were retrieved. Unfilled questionnaires, contradictory answers to questions before and after, and questionnaires with apparent regularity were excluded. A total of 172 valid questionnaires were obtained, with an effective recovery rate of 57.3%. The primary characteristics of the sample enterprises are shown in Table 2.

**Table 2.** Basic Characteristics of Sample Enterprises.

| Variable | Index | Sample Size | Frequency (%) |
|---|---|---|---|
| Enterprise Size | <100 people | 15 | 8.72 |
|  | 100~300 people | 32 | 18.60 |
|  | 300~500 people | 67 | 38.96 |
|  | 500~1000 people | 49 | 28.49 |
|  | >1000 people | 9 | 5.23 |

**Table 2.** *Cont.*

| Variable | Index | Sample Size | Frequency (%) |
|---|---|---|---|
| Years of Establishment | <1 year | 9 | 5.23 |
| | 1~3 years | 27 | 15.70 |
| | 3~8 years | 71 | 41.28 |
| | 9~15 years | 45 | 26.16 |
| | >15 years | 20 | 11.63 |
| Technology Enterprise | Yes | 97 | 56.40 |
| | No | 75 | 43.60 |

### 3.2. Selection and Measurement of Indicators

According to research requirements and index acquisition, the final variable selection and measurement are shown in Table 3.

**Table 3.** Definition and measurement of variables.

| | Variable | Definition | Questionnaire Items | Reference Source |
|---|---|---|---|---|
| Dependent variable | EIP | Enterprise innovation performance. | P1: The degree of the leadership of the company is launching new products.<br>P2: The degree of application of new technologies.<br>P3: Market feedback after product improvement and innovation.<br>P4: The degree of application of advanced technologies.<br>P5: The success rate of new product innovation. | Ritter and Gemunnden [32] Qian et al. [33] |
| Independent variable | NRC | Network relationship Characteristics, including network relationship strength (C1–C3) and network relationship quality (C4–C6). | C1: The frequency of cooperation with upstream and downstream companies.<br>C2: Usually agree with the strategic choices of upstream and downstream companies.<br>C3: Can share resources with upstream and downstream companies.<br><br>C4: Believe in the commitments of upstream and downstream companies and establish long-term cooperative relationships.<br>C5: Satisfaction with the effectiveness of cooperation with supply chain members.<br>C6: Able to consider the overall interests of supply chain members when making decisions. | Granovetter [14] Wu et al. [15] Walter et al. [34] |
| Mediating variable | SCDC | Supply chain dynamic capabilities, including coordination capabilities (S1–S3), learning and absorptive capabilities (S4–S6), and integration and reconstruction capabilities (S7–S8). | S1: From raw material management to production, transportation, and sales, real-time coordination and connection between various departments in the enterprise.<br>S2: Companies can establish a quick order system for major customers, and follow up to receive feedback.<br>S3: Companies can share company demand forecasts and inventory information with major suppliers.<br><br>S4: The ability to grasp the information of the company and its supply chain member companies on time.<br>S5: Regularly train employees and exchange knowledge and experience<br>S6: Ability to integrate new knowledge into an existing knowledge system.<br><br>S7: Identify the challenges and opportunities faced by the company on time.<br>S8: The ability to flexibly adjust business processes between enterprises and departments. | Lin and Peng [8] Yang and Zhu [22] |

**Table 3.** *Cont.*

|  | Variable | Definition | Questionnaire Items | Reference Source |
|---|---|---|---|---|
| Regulated variable | SCGP | Supply chain geographic proximity. | The absolute geographic distance between the company and its largest customer/supplier (1 = absolute distance ≤100 km, 0 = absolute distance >100 km). | |
| Control variables | Size | Enterprise size. | Use the number of employees to measure the size of the company. 1 = Less than 100 people, 2 = 100~300 people, 3 = 300~500 people, 4 = 500~1000 people, 5 = More than 1000 people. | |
| | Years | Years of the establishment. | 1 = The establishment of the enterprise is less than 1 year, 2 = 1~3 years, 3 = 3~8 years, 4 = 9~15 years, 5 = The company has been established for more than 15 years. | |
| | TE | Whether it is a technology-based enterprise. | 1 = Not a technology-based company, 2 = A technology-based company. | |

## 4. Analysis Result

### 4.1. Reliability and Validity Test

Since the research data come from questionnaires, SPSS and AMOS need to be used to verify the authenticity and validity of the data. This article mainly uses SPSS25.0 and AMOS23.0 to analyze the reliability and validity of the scale. The results are shown in Table 4. The Cronbach's $\alpha$ coefficient and the combined reliability (CR) of each variable are above the critical value of 0.7, indicating that the scale has good internal consistency and combined reliability. Exploratory factor analysis of the scale with SPSS25.0 found that the KMO statistics of each variable were between 0.703 and 0.938, all of which were more significant than 0.60 and passed the significance level test of 0.000. Meanwhile, four common factors with more significant characteristics than one were analyzed, explaining 71.168% (more than 50%) of the variance. This indicates that the scale has good structural validity. AMOS23.0 was used for confirmatory factor analysis, and the results showed that the fitting degree of the model was better (x2/df = 1.429; RMSEA = 0.053; CFI = 0.913; TLI = 0.927). Factor loads of all index items were above 0.7, and AVE values of all variables were above 0.5, indicating that the scale had good aggregate validity.

### 4.2. Homology Deviation Test

The study used Harman's unrotated factor analysis method to test the degree of error in the sample data. By analyzing the factors of the variables, the first factor accounted for 27.35% of the total variance (<40%), and thus the homogeneity of the sample data was not serious.

### 4.3. Descriptive Statistics and Correlation Analysis

Pearson correlation coefficient, mean value, and standard deviation of each variable were calculated by SPSS25.0 software, as shown in Table 5. Network relationship characteristics are significantly positively correlated with corporate innovation performance (r = 0.441, $p < 0.01$) and significantly positively correlated with supply chain dynamic capabilities (r = 0.378, $p < 0.01$). Supply chain dynamic capabilities are significantly correlated with corporate innovation performance (r = 0.378, $p < 0.01$). (r = 0.476, $p < 0.001$), and the relevant results provide preliminary support for verifying hypotheses H1 and H2.

**Table 4.** Test Results of Reliability and Validity of Variables.

| Variable | Questionnaire Items | Factor Loading | Cronbach's α | C.R. | AVE | KMO |
|---|---|---|---|---|---|---|
| EIP | P1 | 0.798 | 0.894 | 0.893 | 0.627 | 0.889 |
| | P2 | 0.793 | | | | |
| | P3 | 0.788 | | | | |
| | P4 | 0.796 | | | | |
| | P5 | 0.785 | | | | |
| NRC | C1 | 0.839 | 0.912 | 0.913 | 0.636 | 0.746 |
| | C2 | 0.848 | | | | |
| | C3 | 0.803 | | | | |
| | C4 | 0.763 | | | | |
| | C5 | 0.752 | | | | |
| | C6 | 0.776 | | | | |
| SCDC | S1 | 0.776 | 0.920 | 0.924 | 0.606 | 0.938 |
| | S2 | 0.769 | | | | |
| | S3 | 0.783 | | | | |
| | S4 | 0.766 | | | | |
| | S5 | 0.807 | | | | |
| | S6 | 0.782 | | | | |
| | S7 | 0.781 | | | | |
| | S8 | 0.763 | | | | |

**Table 5.** Correlation Analysis of Variables.

| | Mean | Standard Deviation | EIP | NRC | SCDC | SCGP | Size | Years | TE |
|---|---|---|---|---|---|---|---|---|---|
| EIP | 3.26 | 0.771 | 1 | | | | | | |
| NRC | 3.46 | 0.793 | 0.441 ** | 1 | | | | | |
| SCDC | 3.05 | 0.733 | 0.476 ** | 0.378 ** | 1 | | | | |
| SCGP | 0.41 | 0.494 | 0.314 ** | 0.139 * | 0.403 ** | 1 | | | |
| Size | 3.03 | 1.017 | −0.075 | 0.046 | −0.048 | −0.036 | 1 | | |
| Years | 3.23 | 1.022 | −0.030 | 0.015 | 0.112 | −0.098 | 0.185 * | 1 | |
| TE | 1.37 | 0.483 | 0.048 | 0.016 | 0.051 | 0.024 | 0.300 ** | 0.241 ** | 1 |

Note: ** means significant at the 0.01 level; * means significant at the 0.05 level.

*4.4. Hypothesis Testing*

First, through the simple intermediary model 4 in the SPSS macro program PROCESS, the intermediary effect of the dynamic capabilities of the supply chain between the network relationship characteristics and the innovation performance of SMEs is tested under the control of the size of the enterprise, the age of establishment, and whether it is a technology-based enterprise. As shown in Tables 6 and 7, the results show:

(1) Network relationship characteristics have a significant positive impact on SMEs' innovation performance (B = 0.23, t = 8.63, $p < 0.01$), and hypothesis H1 is valid.

(2) Network relationship characteristics have a significant positive impact on the dynamic capabilities of the supply chain (B = 0.11, t = 6.72, $p < 0.01$), and the dynamic capabilities of the supply chain have a significant positive impact on the innovation performance of SMEs (B = 0.18, t = 6.81, $p < 0.01$). When the intermediary variable of supply chain dynamic capability is put into it, the positive impact of network relationship characteristics on the innovation performance of SMEs is still significant (B = 0.17, t = 9.85, $p < 0.01$). At the same time, the direct effect of network relationship characteristics on the innovation performance of SMEs and the mediating effect of the dynamic capabilities of the supply chain bootstrap 95% confidence interval does not contain 0, indicating that network relationship characteristics can not only positively affect the innovation performance of SMEs, but also the intermediary effect of the dynamic capabilities of the supply chain affects the innovation performance of SMEs. The direct effect and the intermediary effect account for 45% and 55% of the total effect, respectively. Hypothesis H2 holds.

**Table 6.** Intermediary Model Test of the Supply Chain Dynamic Capability.

| Regression Equation (N = 172) | | Fitting Index | | | Significance of the Coefficient | |
|---|---|---|---|---|---|---|
| Outcome Variable | Predictor Variable | R | R² | F | B | t |
| EIP | | 0.29 | 0.08 | 29.35 ** | | |
| | Size | | | | −0.07 | −0.34 |
| | Years | | | | −0.09 | −0.17 |
| | TE | | | | 0.24 | 0.61 |
| | NRC | | | | 0.23 | 8.63 ** |
| SCDC | | 0.37 | 0.15 | 37.23 ** | | |
| | Size | | | | −0.12 | −0.25 |
| | Years | | | | 0.06 | 1.99 * |
| | TE | | | | 0.03 | 0.62 |
| | NRC | | | | 0.11 | 6.72 ** |
| EIP | | 0.40 | 0.16 | 41.23 ** | | |
| | Size | | | | −0.12 | −0.23 |
| | Years | | | | −0.08 | −2.36 * |
| | TE | | | | 0.21 | 0.24 |
| | NRC | | | | 0.17 | 9.85 ** |
| | SCDC | | | | 0.18 | 6.81 ** |

Note: ** means significant at the 0.01 level; * means significant at the 0.05 level.

**Table 7.** Decomposition Table of Total Effect, Direct Effect and Mediation Effect of the Supply Chain Dynamic Capability.

| | Effect Size | Boot Standard Error | Boot LLCI | Boot ULCI | Effect Ratio |
|---|---|---|---|---|---|
| Mediation Effect | 0.28 | 0.04 | 0.19 | 0.38 | 55% |
| Direct Effect | 0.23 | 0.05 | 0.14 | 0.32 | 45% |
| Total Effect | 0.51 | 0.05 | 0.39 | 0.63 | |

Secondly, model 14 in the SPSS macro program PROCESS (consistent with the theoretical model of this study) is used to test the moderated mediation model after controlling variables such as enterprise size, years of establishment, and whether it is a technology enterprise, as shown in Tables 8 and 9. The results show that:

**Table 8.** Moderated mediation model test.

| Regression Equation (N = 172) | | Fitting Index | | | Significance of the Coefficient | |
|---|---|---|---|---|---|---|
| Outcome Variable | Predictor Variable | R | R² | F | B | t |
| SCDC | | 0.39 | 0.16 | 36.33 ** | | |
| | Size | | | | −0.13 | −0.24 |
| | Years | | | | 0.09 | 1.29 |
| | TE | | | | 0.06 | 0.69 |
| | NRC | | | | 0.13 | 6.98 ** |
| EIP | | 0.42 | 0.18 | 43.96 ** | | |
| | Size | | | | −0.11 | −0.21 |
| | Years | | | | −0.07 | −2.33 ** |
| | TE | | | | 0.20 | 0.24 |
| | NRC | | | | 0.16 | 9.79 ** |
| | SCDC | | | | 0.17 | 5.42 ** |
| | SCGP | | | | 0.11 | 4.37 ** |
| | SCDC × SCGP | | | | 0.09 | 6.52 ** |

Note: ** means significant at the 0.01 level; * means significant at the 0.05 level.

**Table 9.** The Moderating Effect of the Geographic Proximity of the Supply Chain.

| Mediating Variable | SCGP | Effect Size | Boot Standard Error | Boot LLCI | Boot ULCI |
|---|---|---|---|---|---|
| SCDC | −0.08 (M − 1SD) | 0.19 | 0.04 | −0.01 | 0.19 |
| | 0.41 (M) | 0.28 | 0.05 | 0.06 | 0.21 |
| | 0.90 (M + 1SD) | 0.32 | 0.05 | 0.07 | 0.28 |

(3) The product term of supply chain dynamic capabilities and supply chain geographic proximity has a significant positive impact on the innovation performance of SMEs (B = 0.09, t = 6.52, *p* < 0.01), indicating that the supply chain geographic proximity is in the supply chain dynamics The relationship between capability and SME innovation performance has a moderating effect. Hypothesis H3 holds.

(4) At the three levels of the geographical proximity of the supply chain, the mediating effect of the dynamic capabilities of the supply chain in the relationship between the network relationship and the innovation performance of SMEs is also regulated (see Table 9). As shown in Figure 2, the higher the level of geographic proximity of the supply chain, the more vital the mediating role of supply chain dynamic capabilities between the network relationship characteristics and the innovation performance of SMEs. Hypothesis H4 holds.

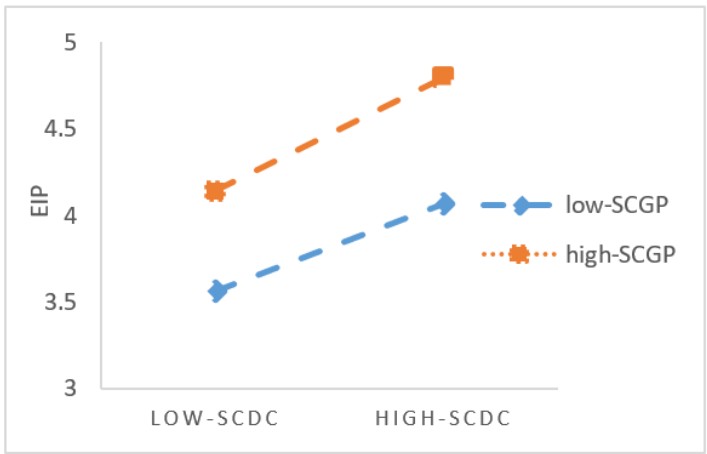

**Figure 2.** The Moderating Effect of Geographic Proximity in the supply chain.

### 4.5. FsQCA Analysis of SMEs Innovation

QCA is a research method based on fuzzy sets and Boolean algebra. It can reflect the nuances of conditional factors, results, and configurations and is suitable for exploring the complex causal relationship between the configuration diversity of conditional factors and specific social phenomena [35,36]. This article selects the QCA method mainly based on two considerations: on the one hand, the innovation of SMEs is the result of multiple factors. On the other hand, in the innovation of SMEs, each antecedent variable and the outcome variable are not necessarily a sufficient and necessary relationship. The traditional statistical analysis method can only deal with the complete corresponding correlation relationship, but the QCA method can deal with various asymmetric relationships. Therefore, this research attempts to combine the set theory model and use the fuzzy set fs QCA method to explore the innovation problems of Chinese SMEs further.

#### 4.5.1. Variable Calibration and Single Factor Necessity Analysis

Regarding the research of Fiss [37], the three calibration points of complete non-subscription, crossover, and entire subordination of each variable are, respectively set as 10%, 50%, and 90% of the sampling frequency of the case. Among them, "SCGP" is

calibrated as Take 0, 0.5, and 1 for the points, respectively, and Take 1, 1.5, and 2 for the "TE" calibration points, respectively. See Table 10 for details.

**Table 10.** Calibration Anchor Points of Variables.

| Variable | Calibration Point | | |
|---|---|---|---|
| | Not Affiliated at All | Intersection | Fully Affiliated |
| NRC | 2.38 | 3.50 | 4.33 |
| SCDC | 2.13 | 2.88 | 4.09 |
| SCGP | 0.00 | 0.50 | 1.00 |
| Size | 2.00 | 3.00 | 4.00 |
| Years | 2.00 | 3.00 | 4.00 |
| TE | 1.00 | 1.50 | 2.00 |
| EIP | 2.26 | 3.10 | 4.40 |

Before the configuration analysis of the antecedent variables, the consistency and coverage of the antecedent variables must be calculated first to assess whether the necessary conditions are available to influence the innovation performance of SMEs. The results are shown in Table 11. The consistency level of each anemic variable in the necessity test of SMEs' innovation performance does not exceed the threshold value of 0.9 (the necessary level identification standard) [38]. It can be considered that no single anemic variable becomes a necessary condition for SMEs to have high innovation performance.

**Table 11.** Necessity Analysis.

| Condition Variable | EIP | |
|---|---|---|
| | Consistency | Coverage |
| NRC | 0.718865 | 0.532849 |
| ~NRC | 0.669862 | 0.568870 |
| SCDC | 0.708666 | 0.686747 |
| ~SCDC | 0.588497 | 0.420601 |
| SCGP | 0.592485 | 0.623834 |
| ~SCGP | 0.416948 | 0.502028 |
| Size | 0.682515 | 0.589404 |
| ~Size | 0.639187 | 0.541234 |
| Years | 0.570936 | 0.626420 |
| ~Years | 0.626227 | 0.508730 |
| TE | 0.495632 | 0.563265 |
| ~TE | 0.536219 | 0.635237 |

Note: ~ refers to logical negation.

### 4.5.2. Empirical Results of FsQCA

Based on the fuzzy set directional comparison and analysis of the corresponding operation rules, the frequency and consistency threshold are set to 2 and 0.8, respectively. The fsQCA3.0 software (Department of Sociology, University of California, Irvine, CA, USA) is used to evaluate the causal adequacy of the innovation performance of SMEs, and the corresponding solutions are obtained. The configuration of factors affecting the innovation performance of SMEs is shown in Table 12. There are three configurations (S1, S2, S3) that produce high innovation performance, and the single solution is consistent with the overall solution. It is higher than the lowest acceptable consistency level of 0.75; the coverage of the overall solution is 0.73, which explains a considerable part of the coverage in the sample cases.

**Table 12.** Innovation performance configuration of SMEs.

|  | S1 | S2 | S3 |
|---|---|---|---|
| NRC | ● | ● |  |
| SCDC | ● | • | ● |
| SCGP | • |  | • |
| Size | ⊗ | ⊗ |  |
| Years |  |  | ⊗ |
| TE |  | • |  |
| consistency | 0.867423 | 0.820017 | 0.798368 |
| raw coverage | 0.402607 | 0.373405 | 0.262653 |
| unique coverage | 0.132285 | 0.184816 | 0.059816 |
| Solution consistency |  | 0.781135 |  |
| Solution coverage |  | 0.733512 |  |

Note: ● indicates the existence of core conditions, • indicates the existence of edge conditions, ⊗ indicates the absence of edge conditions.

(1) Geographical Proximity Adjustment Type S1 (NRC*SCDC*SCGP*~Size). In configuration S1, network relationship characteristics and supply chain dynamic capabilities exist as the core conditions, the geographical proximity of the supply chain exists as an edge condition, and the size of the enterprise is missing as an edge condition.

Type S1 shows that under the adjustment of geographical proximity, the dynamic coordination and integration capabilities of the supply chain of SMEs that lack scale advantages have been effectively improved, and the strong network relationship can promote the innovative behavior of SMEs and improve their innovation performance.

(2) Network Relationship-oriented S2 (NRC*SCDC*~Size*TE). In Configuration S2, network relationship characteristics exist as a core condition, supply chain dynamic capabilities and technology-based enterprises exist as marginal conditions, and enterprise-scale is absent as marginal conditions.

Type S2 shows that technology-based SMEs usually do not have the advantages of scale. Establishing a good relationship with upstream and downstream can strengthen exchanges and interactions and mutual trust between partners. Through the dynamic integration of the supply chain, they can quickly tap resources that are beneficial to their innovation activities. Improve innovation performance.

(3) Dynamically Coordinated and Integrated S3 (SCDC*SCGP*~Years). In Configuration S3, the dynamic capability of the supply chain exists as a core condition, the geographical proximity of the supply chain exists as an edge condition, and the establishment years are missing as an edge condition.

Type S3 indicates that SMEs that have not been established for a long time usually lack the accumulation of network relationships, but when choosing upstream and downstream partners, suppliers/customers with shorter spatial distances can quickly obtain market demand and practical information and promote supply chain dynamics ability to coordinate and integrate innovation resources to improve innovation performance quickly.

4.5.3. Robustness Test

In QCA analysis, robustness testing usually includes multiple methods such as adjusting the calibration threshold, changing the frequency of cases, changing the consistency threshold, adding or eliminating cases [29]. To ensure the robustness of the results, this article conducted a robustness test on the antecedent configuration of the innovation performance of SMEs. First, the consistency threshold was adjusted from 0.8 to 0.75 for configuration construction, and a new configuration was obtained—NRC*SCDC*~SCGP~Size, where NRC exists as a core condition, SCDC exists as an edge condition, SCGP and Size are missing as an edge condition, but the consistency of the new configuration is lower

than 0.75, and the interpretation of the results is that there are no fundamental changes. Secondly, the case frequency threshold was raised from 2 to 3, and it was found that the configuration for realizing the innovation performance of SMEs was the same. Therefore, it can be concluded that the results of this study are robust.

## 5. Conclusions and Suggestions

### 5.1. Conclusions

This study explores the influence mechanism of SME network relationship characteristics on innovation performance under the new dual-cycle development pattern, the mediating role of supply chain dynamic capabilities and the moderating effect of supply chain geographic proximity, and the configuration path of factors affecting SME innovation through fsQCA Carrying out empirical analysis. The following conclusions are obtained:

(1) The characteristics of the network relationship significantly positively affect the innovation performance of SMEs.

The characteristics of network relationships are mainly reflected in two levels of relationship strength and relationship quality. First, the increase in relationship strength can enhance the intimacy of each other's relationship, promote communication and interaction between network partners, and help companies quickly tap favorable resources for innovation; secondly, a good relationship quality can promote trust between network partners, to make the information and resources obtained by the enterprise more valuable, thus avoiding ineffective innovation.

(2) The dynamic capabilities of the supply chain play an intermediary role in the network relationship and the innovation relationship of SMEs.

A good network relationship can directly promote the innovation of SMEs and indirectly promote the innovation activities of SMEs through the intermediary role of the dynamic coordination and integration of the supply chain. Based on a good network relationship between SMEs and network partners, coordinating supply chain dynamic capabilities shared learning, and resource integration can accelerate resource replacement and information acquisition, thereby promoting enterprise innovation.

(3) The different levels of geographic proximity of the supply chain lead to different upstream and downstream coordination and interaction and knowledge sharing efficiency, which positively regulates the relationship between the dynamic capabilities of the supply chain and the innovation performance of SMEs.

The higher the level of geographic proximity of the supply chain, the closer the spatial distance between the company and other network entities, the more convenient it is for the upstream and downstream of the supply chain to coordinate learning and communication and interaction during resource sharing, reduce the asymmetry of information between companies, and thereby reduce the company innovation risk, improve the innovation performance of SMEs. Further through the empirical test of the moderated intermediary model, it is concluded that the geographical proximity of the supply chain not only regulates the relationship between dynamic capabilities and SME innovation performance, but also affects the intermediary effect of supply chain dynamic capabilities between network relationship characteristics and SME innovation. When the distance between the upstream and downstream entities of the supply chain is closer, the intermediary role of the supply chain dynamic capability between the network relationship characteristics and the innovation performance of SMEs is stronger.

(4) There are multiple concurrent causal relationships among the factors affecting the innovation of SMEs. Through the qualitative comparative analysis of fuzzy sets, it is concluded that there are three groups of configurations leading to a high innovation performance of SMEs: Geographical Proximity Adjustment Type (S1), Network Relationship Leading Type (S2), and Dynamic Coordination and Integration Type (S3).

*5.2. Suggestions*

This study provides some valuable insights for SMEs with limited resources, to further rely on network relations to improve innovation performance and achieve sustainable development.

(1) Pay attention to the construction of SME network relations and strengthen external network connections. Network relationships are essential social capital for the development of SMEs. They can provide beneficial high-quality innovation resources for enterprise innovation activities. SMEs should actively participate in the supply chain network, fully tap and utilize the resources in the network, and seize the network. In this way, the innovation performance of enterprises can be improved, and the competitiveness of SMEs can be enhanced.

(2) Strengthen the upstream and downstream cooperation of SMEs to enhance the dynamic capabilities of the supply chain. The dynamic capabilities of the supply chain are conducive to SMEs in quickly responding to turbulent changes in the external environment, seizing innovation opportunities, and improving innovation performance. On the one hand, managers of SMEs can realize resource sharing by strengthening mutual coordination between enterprises and network partners, and on the other hand, by consolidating their foundation and cultivating learning innovation teams to improve the efficiency of the use of innovative resources.

(3) Pay attention to the influence of the geographical proximity of the supply chain, and do an excellent job of screening customers/suppliers. The geographical proximity of the supply chain affects the frequency of upstream and downstream interactions, which is conducive to reducing information asymmetry between enterprises, reducing invalid innovation, and improving innovation efficiency. SMEs managers should choose upstream and downstream customers and suppliers reasonably to ensure that their interests are maximized.

(4) Evaluate the development status of the enterprise and select the innovation mode reasonably. Managers of SMEs should formulate innovation strategies according to their status and development needs. For example, newly established SMEs can quickly obtain innovative resources through dynamic supply chain capabilities. SMEs or science and technology enterprises, which do not dominate in the scale of enterprises, can strengthen the connection between partners by strengthening the network relationship construction and seize the innovation opportunities.

*5.3. Research Limitations*

This study also has some limitations, which need to be further explored in the future.

(1) In the measurement of geographical proximity of the supply chain, due to the lack of reference to existing literature, it is measured by the spatial distance between the enterprise and the first largest customer and between the enterprise and the first largest supplier. Future research can explore the different effects of customer distance and supplier distance on SMEs innovation.

(2) Many factors are influencing SME innovation. In the configuration analysis, only the network relationship, the dynamic capability of the supply chain, geographical proximity level, and enterprise size are considered. In the future, other influencing factors such as industry heterogeneity can be further taken into careful consideration.

(3) Future research can also expand data sources and use panel data and case samples to further verify the research conclusions.

**Author Contributions:** The contribution of all authors was balanced in all phases of the development of this study, both in the empirical part (creation and validation of the instrument, data collection, and analysis) and in the writing part of this manuscript and its various parts. Even the writing of the discussion and the conclusions was produced from a debate among the contributors of the work, which allowed for enriching the arguments based on the different opinions presented. All authors have read and agreed to the published version of the manuscript.

**Funding:** This research received no external funding.

**Institutional Review Board Statement:** Ethical review and approval were waived for this study, as it did not involve personally identifiable or sensitive data.

**Informed Consent Statement:** The study do not involve human nor sensitive data.

**Data Availability Statement:** The data presented in this study are available on request from the corresponding author.

**Conflicts of Interest:** The authors declare no conflict of interest.

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
