# Peer review of "Research on the Influence Mechanism and Configuration Path of Network Relationship Characteristics on SMEs’ Innovation—The Mediating Effect of Supply Chain Dynamic Capability and the Moderating Effect of Geographical Proximity"

_sustainability, doi:10.3390/su13179919_

Round 1

Reviewer 1 Report

The last sentence in the abstract is very long - about eight lines. This can be divided into three or four sentences. There is also a dot after the word “effect” on line 25 that should be apparently eliminated.

In this article, there are many long and non-standard sentences, each of which is more than 6 or seven lines (for example, lines 76-81), and the authors have to re-edit the text and correct the text in this regard. In general, this article should be edited by a native English speaker. There are many incomprehensible sentences that are misspelled. Some of these sentences are very important. For example, lines 222 and 223, which deal with the geographical scope of the study, are: “The research areas include Beijing, Tianjin, Hebei, Shanxi, Guangdong, Jiangsu and other regions”. Or in the previous sentence (from lines 220-222) is very letter-like and it is not clear what concept it intends to convey.

The authors should also note that the citation of the article should be in accordance with the journal citation system.

Unfortunately, there is not a statement of the problem (SOP) in the Introduction. A good introduction explains the problem well and then explains why it is important to address this problem and finally, it should cover the approach this article is taken to address this problem. At the end of the introduction, a paragraph should be devoted to the objectives of the research, detailing what the main purpose and sub-objectives of this article are. Writing a sentence at the end of the introduction alone to express the research objectives is not enough.

This study claims to investigate the impact of network relationship characteristics on innovation in SMEs. However, nowhere in the text of the article is it mentioned what these characteristics are. And it is very important to mention these characteristics.

In the methodology section, it should be mentioned exactly who answered these questions and with what special characteristics, and how they were selected to answer these questionnaires, or in other words, what was the sampling method.

Author Response

Dear review experts:

First of all, thank you very much for your valuable comments on my manuscript.

In response to your comments, I have made the following modifications to the content of the article: 

1.The last sentence in the abstract has been refined into four short sentences. 

2.As the manuscript needs extensive English revision, colleagues majoring in English have been invited to check and revise it. Among them, "the research areas include Beijing, Tianjin, Hebei, Shanxi, Guangdong, Jiangsu and other regions", because the above regions are the gathering areas of small and medium-sized enterprises, and the description has been modified. 220-222 line like letters are the documents on the division standards of small and medium-sized enterprises. The text has been changed and set to an inclined font.

3.The references have been revised according to the Journal Citation requirements.

4.For the modification of the introduction, the description of research questions, research objectives, research methods and research significance are added in the last paragraph of the introduction.

5.The description of network relationship characteristics is described in "2.1. Characteristics of network relationship and SMEs' innovation", but the manuscript does not clearly state that network relationship characteristics refer to network relationship strength and network relationship quality, so the description has been added in the first paragraph of 2.1.

6. The description of sampling method is added in "3.1. Data sources and sample characteristics". The research areas include Beijing, Tianjin, Hebei, Shanxi, Guangdong, Jiangsu and other areas where SMEs gather. The respondents are SMEs that have been established and officially operated for more than 3 years. The questionnaire is filled in by managers who have worked in the enterprise for two years and are familiar with the overall operation of the enterprise.

Reviewer 2 Report

Comments:

This paper presents an analysis of the influence of the social network on the innovation of SMEs. This paper’s topic is interesting, but some aspects need to be improved:

  1. Please check the syntax and punctuation of the whole paper, there are many errors. Es. row 20-21 add “.” not “;” at the end of the phrases; row 25 “fs QCA; row 39 “on the one hand”; row 41 “.”, row 53 “sme”; etc…
  2. Once you define that Small and Medium-size Enterprises are SMEs, please use the acronym in the rest of the paper.
  3. Please check Table 1, use Anglo-Saxon characters.
  4. Please check the Reference format.
  5. Objectives and methods are clear in the abstract, but in the introduction is not clear what you are going to present. What is new in your work? Which methods are you going to use? Why? Will you use a questionnaire? Please also add a presentation of the organization of the paper’s sessions.
  6. Please check rows 135 and 178, the hypothesis presented is one for each session. I also suggest adding a final table to summarized all hypotheses.
  7. Figure 1: Is it a research model?
  8. Table 1 presents the characteristics of the sample, but which are the general characteristics of SMEs? Please introduce this concept concerning the sample.
  9. Why did you choose SPSS 25.0 and AMOS 23.0? Please justify your choice also concerning other works.
  10. Please introduce the statistical methods are you going to use.
  11. Row 377 H1 H2 H3 or S1 S2 S3 like in table 10?
  12. Why did you use fsQCA? Please justify your choice also concerning other works.
  13. A lot of analyses are carried out to validate tests and their results, but a discussion session about results is missing.
  14. The conclusion session should also be further improved to illustrate the practical significance and future research of the study. 

Author Response

Dear review experts:  

First of all, thank you very much for your valuable comments on my manuscript. 

In response to your comments, I have made the following modifications to the contents of the manuscript: 

1. With regard to the grammar and punctuation of the paper, colleagues majoring in English have been asked to help check and revise it.

2. Except for the first definition, "small and medium size enterprises" in the manuscript has been unified with the acronym "SMEs".

3. For the font in Table 1, the Anglo Saxon character is not found and has not been modified. The editor or colleagues will be consulted later.

4. The references have been revised according to the Journal Citation requirements.

5. The description of research questions, research objectives, research methods and research significance has been added in the last paragraph of the introduction.

6. "Hypotheses" in lines 135 and 178 has been changed to "hypotheses", and a table summarizing all assumptions has been added (Table 1).

7. Figure 1 is the research model of this paper.

8. With regard to the general characteristics of small and medium-sized enterprises, the first sentence of the manuscript "3.1. Data sources and sample characteristics" was modified to add a detailed introduction.

9. Why SPSS 25.0 and Amos 23.0 were selected for the study( Relevant works (see Annex 1)

SPSS and Amos were used to analyze the reliability and validity of the scale. The exploratory factor analysis was carried out by SPSS 25.0 to obtain the cranbach's of each variable α The coefficients are above the critical value of 0.7, indicating that the scale has good internal consistency. The kmo statistics of all variables are greater than 0.60 and pass the significance level test of 0.000, indicating that the scale has good structural validity. Amos 23.0 was used for confirmatory factor analysis. The results showed that the fitting degree of the model was better (x2 / DF = 1.429; RMSEA=0.053; CFI=0.913; TLI=0.927;) The factor load of all index items is more than 0.7, and the ave value of each variable is more than 0.5, indicating that the aggregation validity of the scale is good. 

10. Introduction to statistical methods: firstly, 203 data were collected by questionnaire survey for experienced managers of small and medium-sized enterprises, and 172 valid questionnaires were obtained after excluding the questionnaires that were not filled in, contradictory answers to questions before and after, and obvious regularity. Secondly, the reliability and validity of the questionnaire data are analyzed, and the results are good. Secondly, in order to avoid the error in the process of sample collection, the homologous deviation test is carried out. Finally, the questionnaire is analyzed by descriptive statistics, Pearson correlation coefficient, mean and standard deviation.

11. H1, H2 and H3 of article 377 have been modified to S1, S2 and S3.

12. Why fsqca is used (see Annex 2 and Annex 3 for relevant works)

Qualitative comparative analysis (QCA) is a research method between qualitative and quantitative methods, which makes up for the defects of qualitative and quantitative analysis. Based on holism, QCA method believes that there is a complex causal relationship between cases and results. Based on set theory and Boolean algebra, QCA regards cases as conditional configuration, uses conditional configuration instead of independent variables, configuration thought instead of net effect thought, and set relationship instead of correlation, so as to make sociological research enter the era of set analysis from linear analysis.

The QCA method selected in this paper is mainly based on two considerations. On the one hand, SME innovation is the result of multiple factors. From the perspective of enterprise social network, the different combinations of many different antecedents such as network relationship, dynamic capability and network geographical distance form different paths affecting the innovation of small and medium-sized enterprises. Compared with the traditional quantitative analysis method, QCA analysis method can deal with the influence of more antecedent variables on the result variables, so it can better analyze the influence paths of the six antecedent variables in this paper. On the other hand, in the innovation of small and medium-sized enterprises, each antecedent variable and result variable are not necessarily a sufficient and necessary relationship. For example, although the relationship strength in the characteristics of network relationship helps to improve the communication frequency and the speed of resource replacement among enterprises and help small and medium-sized enterprises quickly tap resources beneficial to their innovation activities, it also has the opposite side, Too close relationship strength may rely on network relationship, which is not conducive to innovation. The traditional statistical analysis method can only deal with the completely corresponding correlation. If a → B, then ~ a → ~ B, such as regression analysis and structural equation. Its basic assumptions are independent variables, causal symmetry and one-way linear relationship, while QCA method can deal with all kinds of asymmetric relationships. 

Therefore, this paper adopts QCA analysis method. According to different set forms, QCA analysis methods can be divided into clear set (CS QCA), fuzzy set (FS QCA) and multivalued set (MV QCA). Among them, fsqca can calibrate the analysis conditions with any value between 0 and 1. Most of the variables involved in this paper are continuous variables indicating the degree, that is, each case in this paper is a membership score between 0 and 1. After calibration, the values may be 0.3, 0.6 and 0.7, which goes beyond csqca can only deal with two results of 0 and 1 and the multivalued classification method in mvqca. Therefore, fsqca is more appropriate in this paper. 

13. The discussion on the results has been revised in the section "5.1. Conclusions and discussions". The discussion of fsqca results is mainly added.

14. The conclusion has been improved, and the practical significance and future research direction of the research have been modified accordingly.

Round 2

Reviewer 1 Report

Although all my previous comments are considered in the text, they are not satisfactory. The statement of the problem is provided very poorly in the introduction section and the introduction should be rewritten entirely. 

I still believe that the text requires to be edited by a native English speaker. 

Author Response

Dear reviewer:

Thank you again for your valuable comments on my manuscript.

In response to your comments, I made the following changes to the content of the manuscript:

After sorting out the article, the introduction part has been completely rewritten.

Reviewer 2 Report

Comments:

This paper has been much improved, but further changes are needed.

In particular:

  1. Please check the Reference format in the text.
  2. In the text, I still found “Small and Medium-size Enterprises” instead of SMEs, please check it.
  3. Objectives and methods are now clear in the introduction, but the presentation of the organization of the paper’s sessions is still missing. Es: Session 2 will present the hypotheses of the model, session 3 will explain the model, etc…
  4. Please define the “Innovation Dilemma”; which works use this terminology?
  5. Row 42 is not clear.
  6. Can you add some examples of the benefits of the social network relationship in terms of innovation?
  7. Figure 1: The research model should be implemented. In the figure, I suggest adding for each step input, output, and methods used.
  8. In session three please add a brief description or an example of the questionnaire.
  9. I suggest adding a table in session 3.2 to summarize all indexes presented, which variables they required, and the range of results.
  10. Annexes 1, 2, and 3 are not present in the paper. Please add them and cite them in the text.
  11. Please add a short motivation of the reason why you chose these methods (SPSS 25.0, AMOS 23.0, fsQCA).
  12. I suggest dividing the discussion session from the conclusion and further research session.  

Author Response

Dear reviewer:

Thank you again for your valuable comments on my manuscript.

In response to your comments, I made the following changes to the content of the manuscript: 

  1. The format of references in the text has been checked and revised.
  2. Thank you very much for your careful inspection. The "SMEs" in the manuscript have been revised to SMEs.
  3. Re-organized the introduction and supplemented the presentation of the organization of the paper’s sessions.
  4. "Innovation Dilemma" refers to innovation difficulties, not a professional term. It may be due to the differences in Chinese and English language styles and my own limited level of translation, which has been revised.
  5. Row42: In the process of revising the manuscript, it was found that the description of the problem in the introduction was not clear enough, so it was reorganized.
  6. Examples of the benefits of network relationships in terms of innovation. Added an example in the introduction

Example 1 The Surface 2-in-1 tablet computer developed by Microsoft has promoted the thinner and lighter products of Lenovo and Dell.

In technology-intensive industries such as electronic information and automobiles, the research and development of parts and components and technological innovation are very important and even determine the development of the industry. In the supply chain network, there are many examples of suppliers' independent research and development to promote manufacturer's product innovation. Take Microsoft as an example. As the computer system supplier with the highest market share in the world, Microsoft independently developed a brand-new hardware product-Surface 2-in-1 Tablet PC in 2012. The production of Surface has not only won wide acclaim in the consumer market, it has even led the traditional products of Lenovo, Dell and other computer manufacturers to develop lighter and thinner.

Example 1 BYD and Daimler cooperate in innovation to develop new energy vehicles-DENZA.

In the supply chain network, it is not uncommon for suppliers and manufacturers to cooperate vertically in the research and development of new products. For example, as a battery production supplier, BYD chose to establish a joint venture with the German car company Daimler to jointly produce DENZA new energy vehicles in the form of a corporate alliance when exploring the new energy vehicle market.

  1. According to your suggestion, the hypothesis corresponding to each path is added to the conceptual model Figure 1.
  2. According to your suggestion, the description of the question in the questionnaire has been added to "Table 3 Definition and Measurement of Variables".
  3. According to your suggestion, "Table 3 Variable Definition and Measurement" has been added to 3.2 to summarize all the indicators, which variables they need, and the scope of the results.
  4. Annex 1 is the relevant literature that uses SPSS and AMOS to analyze the reliability and validity of the questionnaire. SPSS 25.0 and AMOS 23.0 are different versions of the software, and there is not much difference.

Annex 2: Qualitative Comparative Analysis (QCA) in Management and Organization Research: Position, Tactics, and Directions. It is about the application of QCA method in management. Already cited in the manuscript

Annex 3 is related literature using QCA method

  1. The brief motivation for using AMOS and SPSS analysis tools has been added in "4.1. Reliability and validity test". The brief motivation for using fsQCA is added in "4.5 FsQCA Analysis of SMEs Innovation".
  2. The discussion part has been separated from the conclusion part and the further research part. The discussion part is added in "4.5.2 Empirical Results of FsQCA".

Round 3

Reviewer 1 Report

Although the authors have considered my remarks and I can see the improvements in the text, especially in the introduction section, I wonder why the authors use the term "my country" not China? Despite the fact that there are two authors in this article so why not our country? I recommend changing "my country" to "China". 

Author Response

Dear reviewer:

Thank you again for your careful examination of my manuscript.

Based on your suggestion, I have changed "my country" to "China" in the article. 

Reviewer 2 Report

This paper has been much improved, I have few suggestions:

  1. Figure 1: The research model should be implemented; it is not so clear in my opinion. In the figure, I suggest adding for each step input, output, and methods used.
  2. I can’t see the table with the questionnaire in the text. 

Author Response

Dear reviewer:

Thank you again for your careful examination of my manuscript.

First, according to your suggestions, and referring to research models of similar works in Sustainability journals, make changes to Figure 1.

Secondly, the questionnaire items of this study are listed in "Table 3 Definition and measurement of variables". In Table 4, only the questionnaire item code is listed before the factor loading of each item, such as P1, P2, P3, etc.